# Bioprocess Economic Modeling: Decision Support Tools for the Development of Stem Cell Therapy Products

**DOI:** 10.3390/bioengineering9120791

**Published:** 2022-12-11

**Authors:** William O. S. Salvador, Inês A. B. Ribeiro, Diogo E. S. Nogueira, Frederico C. Ferreira, Joaquim M. S. Cabral, Carlos A. V. Rodrigues

**Affiliations:** 1Department of Bioengineering, iBB—Institute for Bioengineering and Biosciences, Instituto Superior Técnico, Universidade de Lisboa, Av. Rovisco Pais, 1049-001 Lisboa, Portugal; 2Associate Laboratory, i4HB—Institute for Health and Bioeconomy, Instituto Superior Técnico, Universidade de Lisboa, Av. Rovisco Pais, 1049-001 Lisboa, Portugal

**Keywords:** bioprocess economic modeling, cell and gene therapy, stem cell manufacturing, bioprocess design, quality by design

## Abstract

Over recent years, the field of cell and gene therapy has witnessed rapid growth due to the demonstrated benefits of using living cells as therapeutic agents in a broad range of clinical studies and trials. Bioprocess economic models (BEMs) are fundamental tools for guiding decision-making in bioprocess design, being capable of supporting process optimization and helping to reduce production costs. These tools are particularly important when it comes to guiding manufacturing decisions and increasing the likelihood of market acceptance of cell-based therapies, which are often cost-prohibitive because of high resource and quality control costs. Not only this, but the inherent biological variability of their underlying bioprocesses makes them particularly susceptible to unforeseen costs arising from failed or delayed production batches. The present work reviews important concepts concerning the development of bioprocesses for stem cell therapy products and highlights the valuable role which BEMs can play in this endeavor. Additionally, some theoretical concepts relevant to the building and structuring of BEMs are explored. Finally, a comprehensive review of the existent BEMs so far reported in the scientific literature for stem cell-related bioprocesses is provided to showcase their potential usefulness.

## 1. Introduction

The field of cell and gene therapy (CGT) is currently undergoing a phase of rapid development because of its perceived potential to efficaciously treat all manner of pathologies and even cure hitherto incurable diseases. This perception is well grounded, since the use of living cells as therapeutic agents holds notable advantages over conventional drugs [1]. They can perform a wide range of complex biological functions, such as regenerating tissues, modulating the body’s immune system or actively attacking malignancies, and are capable of adapting to the cues presented by their specific environment. Furthermore, the therapeutic properties of the cells themselves can be enhanced either through cellular hitchhiking or genetic engineering, opening up countless versatile possibilities.

The core concept of CGT is not as novel as it might seem. The first successful instance of cell therapy in a clinical setting occurred as far back as 1956, with an allogeneic bone marrow transplant from a twin donor for the treatment of leukemia [2]. During the following decades, not only did bone marrow transplantation establish itself as an important medical technique with clear therapeutic benefits for several diseases, but other examples of cell therapy also began to appear. In 1997, the FDA approved its first cell therapy product (CTP), intended for the treatment of severe cartilage defects, and this was followed by the approval of several CTPs used as skin replacements to treat burn wounds and ulcers [3].

However, it was not until the discovery of stem cells that CGT truly began to captivate the scientific community. Stem cells, with their inherent ability to self-renew and to differentiate into a variety of cell types, infinitely increased the potential of CGT. On the one hand, they opened up the door to treating diseases associated with extensive loss of cell types which are otherwise difficult to obtain, such as type I diabetes, heart failure or neurodegenerative diseases [4]. On the other, they showed promise in reducing the high production costs and limited efficacy of CTPs reliant on terminally differentiated cells, which are only available in small quantities and possess limited proliferative capacity [5,6]. Aside from hematopoietic stem cells (HSCs), already present in the early bone marrow transplants, mesenchymal stromal cells (MSCs), discovered in the 1970s [7], human embryonic stem cells (hESCs), discovered in 1998 [8], and human-induced pluripotent stem cells (hiPSCs), discovered in 2007 [9,10], have come to be at the forefront of CGT.

Even so, in spite of the advantages inherent to stem cells and the vast efforts dedicated towards the validation of their use for cell therapies, there are not many concrete examples of their application to regular clinical practice. As of 2020, only a total of 10 HSC products have been approved by the competent regulatory agencies [1], along with 10 MSC products [11]. This is in clear contrast with the roughly 300 clinical trials currently ongoing for both HSCs and MSCs [1], as well as the more than 300 trials involving MSCs which have so far been completed [11]. Most notably, no hESC or hiPSC products have been approved as of 2020, although more than 130 clinical trials have been exploring their therapeutic use [12]. In fact, a mere four clinical trials involving hiPSCs have been concluded as of 2021 [13].

The lack of viable CTPs derived from stem cells clearly reveals the difficulty intrinsic to their development. The same biological complexity which makes stem cells so promising for therapeutic applications simultaneously presents an immense barrier to the translation of laboratory-scale experiments into reliable and cost-effective bioprocesses [14,15]. These bioprocesses are invariably intricate affairs composed of a multitude of steps, namely acquisition or generation of the initial cell type, cultivation, modification, harvest, concentration, purification, formulation, fill and finish [14]. Each one of these steps must adhere rigorously to current good manufacturing practices (cGMP) and be optimized to be economically viable. As such, methodical and carefully planned design approaches are indispensable to successfully developing CTPs [16]. This in turn means that tools capable of aiding manufacturers in bioprocess design are unquestionably useful.

In silico bioprocess economic models (BEMs) are a prime example of such tools. In the most basic sense, a bioprocess model is a collection of intertwined equations which represent the various stages of a given bioprocess. These equations receive a series of inputs, which correspond to the raw materials and operational parameters of the stage in question, and convert them into one or more outputs, such as the quantity of end product obtained or the expected quality of said product [17]. To form a BEM, a bioprocess model is overlaid with a layer of economic equations which calculate the cost of the end product based on the consumption of key resources and the production scale. A reliable BEM can be employed to swiftly and inexpensively study how a given bioprocess is influenced by a wide variety of conditions, both from a technical and economic viewpoint. This generates surrogate simulation data for bioprocesses which have yet to be implemented, guiding their process design phase [18] and streamlining process development by reducing experimentation to the bare essential [19]. The conclusions derived from analyzing the results of BEMs can directly inform the decision making of manufacturers, thus making BEMs decision support tools.

The current review seeks to provide a comprehensive overview of the BEMs so far reported in the scientific literature for stem cell-related bioprocesses, in an attempt to illustrate how these models can play an important role in the development of CTPs. In parallel, it presents notions that are fundamental to the design and development of the bioprocesses themselves and explores some theoretical concepts relevant to the building and structuring of BEMs.

## 2. Development of Bioprocesses for Stem Cell Therapy Products

Bioprocesses which aim to manufacture CTPs involve highly variable cell-based raw materials and often fragile final products with short shelf lives [15]. At the same time, they must adhere to rigorous guidelines in order to obtain regulatory approval, which is frequently the source of costly setbacks. Naturally, this leads to laborious and arduous development lifecycles, and there is no assurance that these will result in a viable product suitable for commercialization. As such, it is imperative to follow a strategic framework for cell therapy process development, preferably based on the concept of quality by design (QbD) and its emphasis on continued innovation and iterative refinement, to ensure the success of such ventures.

When developing a CTP with a QbD approach in mind, the first step that must be taken is to carefully define the desired properties of the CTP in question. These are known collectively as the target product profile (TPP) and encompass properties such as cellular identity, potency, purity and quantity [14]. It is equally important to identify which of these properties are absolutely essential for the efficacy and safety of the product, without which it would not exert its function or become a liability to the patient. These are termed critical quality attributes (CQAs). Finally, it is also necessary to ascertain which parameters play a part in determining these attributes, with these being designated as critical process parameters (CPPs). These encompass cellular and noncellular features that range from the specific growth kinetics and gene expression of a cell population to the environmental conditions which influence cell behavior.

As an illustrative example, one of the earliest therapeutic applications of hiPSCs, i.e., the transplantation of autologous hiPSC-derived retinal pigment epithelium (RPE) cell sheets to treat age-related macular degeneration, can be considered [20,21]. In this case, the TPP of the RPE sheets might be said to contemplate the following properties: the expression of typical RPE markers must be above a certain level (identity); the polarized secretion of growth factors associated with native RPE, such as pigment epithelium-derived factor (PEDF) and vascular endothelial growth factor (VEGF), should be significant (potency); the absence of undifferentiated hiPSCs in the final product must be ensured (purity); a sufficient quantity of RPE cells should be produced to form the sheets (quantity). The adequate secretion of PEDF and VEGF clearly constitutes a CQA, since without this, the RPE sheets would not be capable of exerting their biological function. Associated with this CQA is the degree of pigmentation of the hiPSC-derived RPE, which is influenced by a number of CPPs, such as the concentration of the growth factors used to induce the differentiation of hiPSCs toward RPE cells and the duration of the differentiation process.

Evidently, the TPP of a CTP cannot be fully defined during the early stages of its development, given that there is limited knowledge available regarding how the product will really act in a clinical setting [15]. This is unavoidable, and thus the concept of iterative refinement is at the core of QbD. The idea is that, after roughly outlining the TPP and CQAs, a process design space should be defined, representing the many parameters which interact in such a way as to affect the CQAs. The TPP should then be iteratively refined through the accrual of understanding about the product’s mode of action, derived from clinical trials, and the CPPs, derived from extensive monitoring and collection of experimental data. This refinement is reflected in the gradual narrowing of the design space as process optimization becomes focused on a smaller number of parameters.

Iterative refinement can best be guided by taking advantage of two complementary strategies, namely design of experiments (DOE) and computational modeling [14,15]. DOE comprises a set of methodologies that attempt to evaluate multiple factors simultaneously so as to understand the impact of their interactions on the output of a given bioprocess [22]. The data obtained from these meticulously designed experiments can afterwards be used to build a computational model which describes the expected behavior of the bioprocess as a function of certain input parameters. Such a model might be limited to technical aspects or expanded to also include economic considerations, as is the case of BEMs. The model can then be employed to explore and fine-tune the design space in silico, identifying which parameters are most promising for optimization and streamlining further experimental process development [15]. The QbD framework, evidencing the roles which DOE and bioprocess economic modeling play in its implementation, is represented schematically in Figure 1.

Returning to the example of the RPE cell sheets, it is easy to see how a BEM could help in the optimization of this CTP. As was mentioned, the purity of the final product, i.e., the absence of undifferentiated hiPSCs that could prove to be tumorigenic, is a CQA. In the initial production strategy, this purity was guaranteed solely through a series of manual “pick up and expand” procedures, whereby cells which visually resembled the desired RPE cells were selectively expanded over several passages before the formulation of the final product [20]. Not only is this procedure extremely time-consuming, but it also requires highly specialized personnel. Therefore, a manufacturer would likely consider improving this step of the bioprocess by implementing an affinity purification strategy. After some consideration, he may come to the conclusion that he should choose either fluorescent-activated cell sorting (FACS), magnetic-activated cell sorting (MACS) or buoyancy-activated cell sorting (BACS). But which would be best? After some preliminary experiments, carried out with the objective of determining the yield and purity of the strategies under consideration, the manufacturer might arrive at the conclusion that FACS and MACS allow for a somewhat higher purity than BACS, but that BACS is significantly cheaper to implement due to reduced labor and equipment costs. So the question remains, which should he choose? This decision would ideally be made by employing a BEM to compare the cost-effectiveness of each strategy when integrated within the overall bioprocess. The BEM could estimate, for example, that executing two or three consecutive BACS purification steps would lead to a purity comparable to that of FACS and MACS while still being less expensive, thus giving the manufacturer a solid reason for choosing BACS in detriment of the other strategies. Similar observations could be made about the many other decisions the manufacturer would be confronted with, such as selecting the ideal hiPSC reprogramming strategy or the optimal retinal culture medium.

Interestingly, computational models should undergo iterative refinement alongside their respective bioprocesses, since, as more becomes known about an underlying bioprocess, the robustness and fidelity of its respective model can be progressively increased. Recent work by Manstein et al. presents an excellent example of this [23,24]. In their work, after establishing an initial mechanistic model based on experimental data, they cyclically challenged the model by altering input parameters and verifying if the model’s outputs corresponded with reality. When they did not, the necessary adjustments were made to bring the model closer in line with the observed experimental results. This allowed a model with strong predictive capability to be built, which was then successfully used to optimize culture conditions and maximize hiPSC density in stirred-tank bioreactors.

Another fundamental aspect heavily encouraged by QbD is continuous process innovation. This means that, whenever favorable, novel manufacturing strategies and elements with the potential to either reduce bioprocess complexity or increase product quality should be adopted [15]. A prime example of this is the transition from planar culture formats, ubiquitous in laboratory settings and thus likely to be used in the early stages of development, to automated bioreactor culture, so as to benefit from improved scalability, integrated manufacturing and increased cell quality [25,26]. Other sources of innovation include the development of less costly and more rigorously defined culture media [27], ingenious cell purification methods [28] and streamlined cell characterization techniques [29]. As will be seen further in this review, BEMs are particularly well suited to identifying whether a given innovation should or not be implemented, both from a technical and economic perspective. This is important in order to justify the large investment of time and monetary resources which normally accompanies innovation.

As a last note concerning QbD, the development of bioprocesses which are composed of multiple sequential stages should give rise to modular manufacturing processes [15]. In other words, these bioprocesses can be organized as several modular blocks, each encompassing a certain set of unit operations at the end of which an intermediate cell population is obtained, with its own distinct TPP and CQAs. This allows for a much simpler optimization of the overall bioprocess, given that each modular block can be individually optimized without interfering with another. Notably, this same modularity should be reflected in any eventual BEM, so that it can be used to independently explore the design space of each modular block.

Once more looking to the RPE cell sheets, the applicability of this modular approach is clearly evident [20,21]. Fibroblasts from a patient must first be reprogrammed into hiPSCs, followed by their differentiation into RPE cells and then their expansion until enough cells are obtained to form sheets. At the end of the reprogramming block, an intermediate cell population of fully characterized, pluripotent and genetically stable hiPSCs is desired. On the other hand, upon completion of the differentiation block, an intermediate cell population of aptly differentiated, RPE marker expressing and PEDF and VEGF secreting cells is expected. The CPPs that influence the CQAs of the cell populations of each of these blocks are indubitably distinct, and can therefore be iteratively refined separately within their own specific design spaces, decreasing the complexity of this endeavor.

## 3. Characterization of Bioprocess Models

Bioprocess models can be classified according to a number of different characteristics, usually related with the complexity and realism of the model itself. In general terms, more intricate models are imbued with a greater degree of fidelity, and therefore more robustly recapitulate the bioprocess they attempt to represent. Commonly, models are considered to be either static or dynamic, deterministic or stochastic and mechanistic or data-driven [17,18]. These notions are explored in detail in the following subsections in order to better understand the structure of bioprocess models and to facilitate the interpretation of their results. Table 1 summarizes the main advantages and disadvantages of each one of the covered model classifications.

### 3.1. Static versus Dynamic Models

Static models are relatively simple and easy to build, being an invaluable tool for cost estimation at the earlier stages of process development [17,30]. They can be used to test different sets of inputs and process parameters to assess their impact on process costs, allowing for the straightforward comparison of different scenarios. The major drawback of these models is that they are incapable of capturing complex logistical operations that are dependent on time and resource constraints. These can often arise in the context of CTP bioprocesses, which involve a wide variety of lengthy procedures, parallel processing and limited equipment availability, meaning that the estimates provided by static models can be expected to deviate somewhat from reality.

Conversely, dynamic models are more complex and designed with the express purpose of integrating time-dependent operations and discrete-event simulation [17,30]. Each production batch is simulated individually, with the system state being updated throughout the simulation’s timeline as each batch progresses through consecutive events (e.g., expansion, differentiation, downstream processing, etc.). Several batches can be simulated simultaneously, making it possible to represent parallel processing and facilitating the identification of potential bottlenecks. This means that dynamic models are particularly useful for optimizing factors such as the number of operators or units of equipment that are necessary for a bioprocess to run smoothly at a given production scale. The intricacy of the simulations that underlie dynamic models can sometimes make them difficult to interpret, but their increased fidelity means that their estimates are often quite close to reality.

### 3.2. Deterministic versus Stochastic Models

Deterministic models present their outputs as definite outcomes and do not take into account the risk that accompanies variable input parameters and the impact of these variations on a model’s outputs [17]. This is particularly disadvantageous in the case of bioprocess modeling because of the inherent biological variability of cell-based raw materials. This results in variable process yields and processing times, along with batch failures that can arise from unpredicted cellular events or culture contamination [14]. Risk-based analysis is therefore of paramount importance. Contrary to deterministic models, stochastic models determine a range of outputs by incorporating the manufacturing uncertainty that characterizes bioprocesses into their simulations, thus being more suitable for describing them.

Three methods that can be employed to incorporate uncertainty into process models are case-based modeling, risk adjustment and Monte Carlo simulation [31]. In case-based modeling, a set of values is defined for each input and the outputs for different case scenarios are computed. Typically, outputs for a best, baseline and worst case scenario are evaluated [17,31]. In risk adjustment modeling, a set of inputs, each weighted by their likelihood of occurrence, are used to generate several outputs. Expected averages are then calculated from all of the possible outcomes, originating risk-adjusted values as the final outputs [17]. Monte Carlo simulation is the most statistically robust of the mentioned methods and is commonly employed for stochastic modeling whenever possible. This method is implemented by establishing probability distributions for a model’s inputs, based either on experimental data or subjective estimates from experts. A large number of simulation runs are then performed by randomly sampling values from the established probability distributions, with the samples of each run being used to determine their respective outputs. This results in the generation of output probability distributions, allowing not only to identify the range of possible outcomes but also to analyze the variability of each output and the probability of exceeding a certain critical threshold [17,31,32].

### 3.3. Mechanistic versus Data-Driven Models

Mechanistic models, which can be subdivided into kinetics-based and flux-based models, rely on first principle mechanisms to describe process behavior [18]. As such, a substantial amount of knowledge concerning the process being modeled is required, along with an extensive collection of data that can be used for parameter estimation and validation. Kinetics-based models typically use systems of ordinary or partial differential equations to describe observed phenomena and to accurately predict process outcomes. An example of this is the modeling of cell metabolism dynamics through variations of the Monod equation. On the other hand, flux-based models describe steady-state genome-level cell metabolism dynamics, being suitable for exploring how metabolic pathways can be modulated to optimize the yield of the desired final metabolite(s). It is important to note that, in reality, the majority of mechanistic models tend to be of a semimechanistic nature, given that the impact of some process inputs on their respective outputs is unknown. For example, when complex culture media formulations are used, the exact influence of each specific media component on cellular metabolic pathways is not necessarily understood. Because of this, unknown mechanisms are often remedied with empirical approximations.

As opposed to mechanistic models, data-driven models exclusively base themselves on experimental data to describe the relationships between process inputs and outputs. In other words, a data-driven model estimates that a point B should be reached if starting from a point A, but it does not attempt to describe the path taken to do so. While these models do not explain observed phenomena, they are very much invaluable, since they can capture important process information in a straightforward way. Additionally, they are useful for modeling processes whose fundamental mechanisms are not well understood but for which some experimental data exists, which for most CTP bioprocesses is likely to be the case [18].

## 4. Bioprocess Economics

Insofar as bioprocess economics are concerned, two key cost-metrics must be reckoned with for the estimation of manufacturing costs, namely fixed capital investment (FCI) and cost of goods (CoG) [33]. A firm grasp of these metrics is indispensable in order to integrate the appropriate cost equations within a bioprocess model. Another concept that is particularly important for determining the expenses incurred by a bioprocess whose end product is intended for therapeutic applications is cost of quality (CoQ). This is because the quality of these products must be categorically assured in accordance with strict cGMP guidelines, which are naturally associated with hefty costs [25].

### 4.1. Fixed Capital Investment and Cost of Goods

The FCI represents the total initial investment required to build a manufacturing facility with all of the necessary installations and equipment to execute a given bioprocess. Aside from depending directly on the manufacturing structure of the bioprocess itself, the value of the FCI can be influenced by several factors, such as the ratio between open and closed cleanroom areas, the choice between automated versus manual processing and the use of disposable or nondisposable process platforms [33]. To determine the total initial investment involved in the establishment of a manufacturing facility for stem cell processing, two models can be implemented: a cost-per-area-model or a Lang factor model [32]. The first model consists in defining the cleanroom area of a facility along with its noncleanroom area, summing both areas multiplied by their respective price-per-square-meter (or another unit of area) and then adding the total equipment acquisition cost [33]. The Lang factor model determines the FCI by multiplying the total equipment acquisition cost by a cost factor, which should be based on the relationship between the FCI and equipment acquisition cost observed for past projects of a similar nature [32,33].

The CoG incorporates all direct and indirect costs that arise from the manufacture of a CTP [33]. Generally speaking, the CoG is expressed in relation to a single therapeutic dose. Direct costs can be further broken down into three categories, namely consumable, reagent and quality control costs [32]. Consumables include all disposable lab equipment, such as single-use vessels, serological pipettes, gloves, etc. Reagents comprise all of the chemical or biological substances used throughout a bioprocess, such as cell culture medium, dissociation enzymes, buffers for product storage and final formulation, etc. Quality controls encompass any procedure carried out for intermediate or final CTP testing, which seek to ensure that it satisfactorily fulfills its CQAs. These procedures are thus highly specific to each bioprocess.

Indirect costs can broadly be subdivided into infrastructure and labor costs [32]. Infrastructure costs cover a wide variety of expenses, including facility and equipment depreciation. Depreciation denotes the loss of an asset’s value with time, which is often considered to occur in a linear manner [34]. In the case of a facility and its equipment, the depreciation rate of each constituent is equivalent to its associated portion of the FCI divided by its respective lifetime, with the lifetime of the facility generally being longer than that of its equipment. Lengthy bioprocesses that occupy a facility and its equipment for a long time during each production batch will naturally originate higher depreciation costs than shorter bioprocesses. Infrastructure costs also encompass operating costs, such as those incurred by the supply of gases (O_2_, CO_2_, N_2_), cleaning and maintenance of the facility, requalification procedures, etc., storage costs and transportation costs. These last two are heavily influenced by the final formulation of the product in question, with cryogenically preserved products almost always being cheaper alternatives to fresh-preserved products, due to their prolonged shelf life [35]. This means that in some cases it may be advantageous to preserve a CTP at an intermediate stage if this allows it to be cryopreserved, with its final formulation being completed before its delivery to the site of care. Labor costs include salaries, benefits and expenses associated with the specialized training of personnel.

### 4.2. Cost of Quality

As has been repeatedly mentioned, CTP bioprocess design and development is a challenging affair that carries many risks. Manufacturing failures can arise from a multitude of factors, such as the biological variability inherent to cell-based raw materials, the high sensitivity of cells to their physiochemical environment, which might lead to spontaneous differentiation or unpredicted cell death, and culture contaminations. These manufacturing failures can have a large financial impact, preventing products from reaching the market. Furthermore, regulatory agencies have stringent requirements in terms of process manufacturing and control, which, when not met, can result in products being held back from regulatory approval. Therefore, bioprocesses should be carefully designed to achieve the desired high productivity and product quality, while ensuring their economic viability [14,18,25,36].

CoQ models are of key importance for identifying and measuring the costs associated with quality assurance, helping to minimize these costs while maintaining a certain level of quality [37]. CoQ takes into account the costs of preventing poor quality along with the costs incurred by the nonconformance with quality criteria that causes manufacturing failures. The majority of CoQ models follow a prevention-appraisal-failure (P-A-F) scheme. These models divide quality costs into prevention costs, which are associated with actions taken to ensure the quality of a product, appraisal costs, related to measuring the level of quality of a product, and failure costs, corresponding to the costs of correcting the quality of a product. The main principle of P-A-F models is that investing more in prevention and appraisal activities frequently proves to be beneficial by significantly decreasing failure costs. This is particularly true when talking about bioprocesses since their end product’s quality is rarely correctable, meaning that when a batch failure occurs the resources invested towards its production are entirely wasted and a new batch must be started to replace the lost product.

## 5. Review of Stem Cell-Related Bioprocess Economic Models

Given the inherent advantages of bioprocess economic modeling, it comes as no surprise that the development and application of BEMs has garnered significant academic interest. Traditionally, these have mostly been applied to biopharmaceutical protein production, among which antibody production can be highlighted [17]. Only more recently has attention also begun to be placed upon other end products, such as those encompassed by the up-and-coming CGT sector. Some prominent examples of these are lentiviral vectors [38], CAR-T cells [39,40], and undifferentiated or differentiated stem cells. Because of this, the design of BEMs oriented towards this sector is still in its infancy, although several pioneering works have already presented significant contributions and laid out important foundations.

Despite the valuable technical insights which can be gleaned from studying bioprocess models from distinct areas of study, the current section is focused on providing a comprehensive review of those specifically concerned with stem cells, namely HSCs, MSCs, hESCs and hiPSCs, or their derivatives. A summary of the reviewed works is presented in chronological order in Table 2 and their characteristics and subject matter are discussed in greater detail in the following subsections. The discussion begins with an overview of the biological aspects that must be taken into consideration when designing a bioprocess (and its respective BEM) for specific stem cell types. This is followed by short descriptions of each of the reviewed works, organized according to the main purpose behind the BEMs that they present. The discussion then finishes with an overview of the technical aspects of the covered BEMs. This discussion intends to showcase the very wide range of applications of BEMs in the context of CTP bioprocessing without going into too much detail, with readers thus being encouraged to explore more deeply those works that cover topics pertinent to their own interests.

### 5.1. Overview of Cell Types, Their Origin and Their Intended Applications

When analyzing the available literature concerning stem cell-related BEMs, a key consideration to keep in mind is that the choice of which stem cell type should be employed for a given bioprocess is intimately linked to the intended application of its respective CTP. HSCs are most commonly administered as a single cell-suspension in their multipotent state to counteract disorders which affect the hematopoietic system [1]. Alternatively, they can first be differentiated in vitro into red blood cells (RBCs), which are then suitable for traditional transfusion medicine [58]. Aside from this, RBCs can act as particularly efficient carriers for delivering therapeutic agents to target sites, combining nanomedicine with cell therapy [59]. Similarly to HSCs, MSCs are also frequently administered in their multipotent state, either to benefit from their immunomodulatory properties to treat autoimmune disorders [60], or to take advantage of their ability to naturally differentiate into the cell types relevant to the tissue being treated while simultaneously promoting endogenous repair processes through their paracrine activity [61].

In contrast, hESCs and hiPSCs must be thoroughly differentiated before being administered to a patient, since residual hESCs or hiPSCs carry a high risk of tumorigenicity [62]. As such, cell therapies involving hESCs and hiPSCs do not rely on them directly, but rather on their differentiated descendants, which are usually organized as cell sheets or embedded in scaffolds before transplantation [63]. This notwithstanding, it should be noted that, because of their pluripotent nature, hESCs and hiPSCs can differentiate into any cell type, including HSCs or MSCs, making them the most versatile of the stem cells so far described. Because of this, hESCs or hiPSCs are invaluable cell-based raw materials for producing CTPs which involve cell types that are relatively inaccessible under normal circumstances, such as cardiomyocytes, hepatocytes or neurons.

Another very important aspect to consider is whether the CTP in question is autologous or allogeneic in origin. Autologous CTPs are produced from cells belonging to the individual being treated, while allogeneic CTPs are derived from especially recruited donors which are distinct from the target patient. This has an important impact on the scalability of the bioprocess and its associated costs [36]. Allogeneic CTPs can benefit from economies of scale due to manufacture scale-up, with a single batch being used to produce many patient doses. Contrariwise, autologous CTPs must necessarily be scaled-out as demand increases, given that each batch produces cells to treat a single patient. Thus, allogeneic CTPs often benefit from reduced labor and operating costs. Another challenge incurred by autologous CTPs is donor variability, which hampers process characterization and must be well understood to guarantee product quality. Allogeneic CTPs avoid this challenge by always resorting to cells from the same donor pool, facilitating process standardization.

This of course means that the structure of the bioprocess which gives rise to a certain CTP can vary significantly according to the stem cells employed and their origin. CTPs derived from hESCs and hiPSCs are often associated with higher production costs and face greater difficulty when it comes to having their quality categorically assured. HSCs and MSCs can be collected directly from patients, whereas hESCs require the destruction of human blastocysts, being associated with ethical concerns [12], and hiPSCs are derived from laborious reprogramming processes with very low efficiency, which can give rise to unforeseen genetic and epigenetic alterations in cells when not strictly controlled [64]. Furthermore, the tumorigenic potential of hESCs and hiPSCs demands extremely robust differentiation and purification protocols [62], significantly increasing the complexity of these products.

Out of the 19 BEMs reviewed in this work, 11 employ MSCs as cell-based raw materials, 5 use hiPSCs, 2 use HSCs and 1 uses hESCs. Nearly all of the BEMs involving MSCs sought to simulate the manufacture of allogeneic CTPs, while those involving hiPSCs often sought to simulate the production of autologous CTPs. This trend can be explained by the historical, although not entirely correct, tendency to consider MSCs as immune privileged [65]. By avoiding the harsh immune response normally caused by the transplantation of allogeneic cells, MSCs become tempting targets for the perceived economic benefits of allogeneic CTPs. On the other hand, terminally differentiated hiPSCs do not benefit from any immune privilege, and the potential for them to be readily derived from a patient’s own cells makes them especially compatible with autologous CTPs [63].

Overall, it seems clear that CTPs based on hESCs or hiPSCs are at the present moment less economically viable than those based on HSCs or MSCs. This perspective is corroborated by the models covered in this review and the fact that there already exist some HSC and MSC CTPs, but no hESC or hiPSC CTPs have as of yet been commercialized. In large part, this can be attributed to the contrast which exists between autologous hESC or hiPSC therapies and allogeneic HSC or MSC therapies. The latter can much more easily benefit from economies of scale, as well as rely on the selection of a small group of ideal donors, whose cells are particularly easy to expand or show increased therapeutic potential. The careful selection of donors can significantly decrease the cost of a given bioprocess by reducing the time taken to achieve its target cell number, which concomitantly minimizes operating costs and the consumption of key resources, as evidenced by some of the BEMs here reviewed [35,53]. Moreover, the added complexities of maintaining and processing hESCs and hiPSCs, versus HSCs and MSCs, translate into higher quality control and personnel qualification costs, increasing expenses even further.

### 5.2. Review of Models Used for the Optimization of Bioprocess Design

Many of the reviewed BEMs were developed with the intent of determining the optimal design of the bioprocess being modeled. Most of these sought to optimize the configuration of cell expansion, differentiation, harvesting, purification, processing or fill finish technologies, focusing on only one of these aspects or on several at once [41,42,44,47,49,50,56]. This can be seen as a primarily macroscopic approach geared towards global bioprocess design. Distinctively, a couple of the works presented in this review, namely those concerned with the production of RBCs from HSCs, directed their attention to the optimization of cell culture conditions in a specific technology, which can be considered a more microscopic approach [48,52].

Simaria et al. developed a BEM to simulate the large-scale expansion of allogeneic MSCs in the context of cell therapy manufacturing [41]. The model was used to evaluate the technical and economic competitiveness of several cell expansion technologies across a wide range of production scales. Several types of planar technologies, both manual and automated, as well as single-use bioreactors, were evaluated. The model identified the optimal cell expansion technology, along with its size and amount, for each production scale. This was done by computing the cost of all of the possible technology configurations and choosing the one which was capable of fulfilling the specified production target while minimizing the CoG. In this way, the competitiveness of each technology was ascertained and their limits were delineated. In general, compact planar technologies were found to be optimal for small to intermediate production scales, after which bioreactors become indispensable. A sensitivity analysis carried out with the model also explored the influence of several bioprocess parameters on its cost outcome. Of these, microcarrier area and harvest density seemed to be the most relevant for future improvement.

Darkins and coworkers reported a BEM oriented toward the large-scale production of allogeneic hiPSC-derived cardiomyocytes [42]. It was focused on the optimization of the differentiation, harvesting and purification portions of this bioprocess, which sought to produce between $10^{14}$ and $10^{15}$ cardiomyocytes per year for regenerative medicine applications. The model was developed by combining biomechatronic analysis with a computer-aided design approach. Four distinct bioreactor configurations, along with planar technologies, were evaluated for differentiation, while eight techniques were examined for harvesting and purification. When it comes to differentiating hiPSCs into cardiomyocytes, wave bioreactors were found to be the most ideal technology. In terms of harvesting and purifying the produced cardiomyocytes, a combination of density gradient centrifugation, filtration and aqueous two-phase system separation was determined to be optimal. The reduced capital investment and operating cost of the chosen technologies were the main reasons for them being perceived as advantageous in comparison to the other technologies which were evaluated. Based on these results, a detailed bioprocess was designed using a computer-aided design software and it was estimated that its payback period would be 4.7 years.

Hassan et al. further developed the BEM published by Simaria et al., once again simulating the large-scale expansion of allogeneic MSCs for the same range of production scales [44]. However, this time focus was not placed on cell expansion technologies but rather on downstream processing and fill finish technologies. Different sizes of tangential flow filtration (TFF) and fluidized bed centrifugation (FBC) systems were evaluated for downstream processing, along with vial filling technologies with distinct filling capacities for fill finish. The expanded model identified the optimal downstream processing and vial filling technology for each production scale. To accomplish this the cost of all of the potential technology configurations was calculated and the one which was capable of fulfilling the specified production target while minimizing the CoG was selected. It was found that TFF is more cost-effective for small to intermediate production scales while FBC becomes the only option for larger production scales. Furthermore, a lack of downstream processing technologies appropriate for the largest production scales was clearly identified, indicating that further improvement of the existing technologies was required, coupled with better process logistics, such as staggered upstream processing.

Weil and coworkers established a BEM for simulating the manufacture of autologous hiPSC-derived progenitor photoreceptors [47]. Its main concern was the technoeconomic optimization of the affinity purification technology employed at the end of this bioprocess. Three technologies were compared, namely fluorescent-activated cell sorting (FACS), magnetic activated cell sorting (MACS) and SpheriTech beads, for several production scales. Purification was shown to account for only roughly 10% of the total CoG, while differentiation accounted for approximately 50%. Despite this, the technical performance of the selected purification technologies had a significant retroactive impact on differentiation costs, since the number of cells required at the end of differentiation depends directly on the purification yield. FACS was found to be economically favorable for the smaller production scales, whereas MACS clearly outperformed the other technologies for larger scales. SpheriTech beads were incapable of competing with the better established FACS and MACS mostly because of their relatively poor yield. This notwithstanding, it was determined that if their yield were to be at least mildly improved, SpheriTech beads could readily present themselves as the superior alternative.

Misener et al. developed a BEM which accurately predicted erythropoietic maturation of RBCs derived from HSCs in a dual hollow-fiber bioreactor [48]. The model was geared towards the optimization of bioreactor design and operation, combining several complex mathematical tools, such as stem cell fate modeling, computational fluid dynamics and superstructure optimization, for this purpose. It was determined that an optimal bioreactor design would lead to a cost of $277 per unit of RBCs, reducing the production cost by a factor of 4 comparatively to the bioreactor design before optimization. A more robust solution, inoculated against uncertainty, would lead to a cost of $383 per unit of RBCs. Even so, the results obtained with the bioreactor compared very favorably with traditional 2D static culture from an economic perspective. A single variate and multivariate sensitivity analysis carried out with the model indicated that cellular flux, the half-life of species relevant to RBC production, such as stem cell factor and erythropoietin, and cellular kinetic parameters had the greatest impact on bioprocess cost.

Chilima and coworkers reported a BEM designed to evaluate the operational and economic performance of four cell expansion technologies for the manufacture of allogeneic MSCs at various production scales [49]. Three planar technologies were evaluated, namely multilayer flasks, multiplate bioreactors and hollow-fiber bioreactors, along with stirred-tank bioreactors. The optimal technology was chosen for each scale not only based on a direct comparison of CoG but also on multiattribute decision making, which additionally weighs operational characteristics, such as ease of development, validation and setup. Multiplate bioreactors were shown to be advantageous at small scales, while stirred-tank bioreactors become the superior alternative at intermediate and large scales. Notably, multiplate bioreactors clearly outperformed the other two planar technologies, and even for the larger production scales closely competed with stirred-tank bioreactors. This observation was attributed to their superior operational characteristics. Moreover, a sensitivity analysis undertaken with the model showed that the processing capacity of its contemplated downstream processing technology, an FBC system, was a major limiting factor for large-scale production, thus requiring significant improvement.

Mizukami et al. presented a BEM with the intent of studying the large-scale expansion of MSCs derived from umbilical cord matrix for the treatment of acute graft-versus-host disease [50]. The model was adapted from the work of Chilima and colleagues, which was altered to include different performance parameters based on experimental results. The evaluated technologies were multilayer vessels, stirred-tank bioreactors, hollow-fiber bioreactors and packed-bed bioreactors. Multilayer vessels were found to be the most cost-effective technology, in spite of hollow-fiber bioreactors presenting the best experimental results. In fact, hollow-fiber bioreactors were the least cost-effective of the four technologies. This exemplifies perfectly the role of BEMs, which are essential for showing that a potential technology is not only technically adequate but also economically viable. A reimbursement analysis carried out with the results of the model indicated that none of the contemplated technologies would be capable of achieving an acceptable CoG as percentage of sales for a selling price of $25,000, thus identifying the need for further optimization of the bioprocess and its technologies.

Glen and coworkers established a BEM representative of erythroblast growth in stirred-tank bioreactors, whose purpose was the economic optimization of this same bioprocess [52]. The main feature of the model is its ability to accurately describe an experimentally observed phenomenon of erythroblast growth inhibition through a hypothesized autocrine feedback loop involving the accumulation of an unknown inhibitory factor in the culture medium. This allowed the model to be used to determine the optimal cell density and medium exchange time, which maximized production efficiency by mitigating the inhibitory effect. The production efficiency was calculated taking into account time-dependent operating costs and culture medium costs, which were considered to be the main cost drivers. The optimal exchange time occurred earlier for smaller production scales, since for these, the culture medium has a smaller impact on total bioprocess cost. It was estimated that to produce enough erythroblasts for a unit of RBCs ($2\times 10^{12}$ cells) at a price of $5000 the cost of the culture medium should not exceed $4/L and the operating costs should be no more than $30/h.

Ng et al. reported a BEM for studying the large-scale production of extracellular vesicles (EVs) derived from allogeneic MSCs for therapeutic applications [56]. With this model, the economic competitiveness of different combinations of cell expansion and EV harvest technologies was evaluated across a wide range of production scales. Numerous types of planar platforms and single-use bioreactors were analyzed for cell expansion, while ultracentrifugation, polymer-induced precipitation, size-exclusion chromatography and ultrafiltration were analyzed for EV harvest. The model identified the most favorable combination of technologies by computing their associated cost and choosing the combination which fulfilled the imposed restrictions while minimizing the CoG. Harvesting of EVs was found to be a major contributor to bioprocess costs, often accounting for more than 50% of the annual CoG. Ultrafiltration was shown to be the most cost-efficient technology for the majority of production scales, being capable of reducing the cost contribution of EV harvest to as little as 20% in some cases. A sensitivity analysis performed with the model indicated that biological parameters, namely the rate of EV production by a patient’s cells, are the most significant cost drivers.

### 5.3. Review of Models Used for the Comparison of 2D and 3D Expansion Technologies

As of recent years, bioreactors have increasingly come to be viewed as a superior alternative to traditional planar platforms when it comes to consistent and economically viable stem cell culture. On the one hand, bioreactors are capable of more closely reproducing the in vivo environment in which cells are naturally extant [66]. On the other, the fact that a bioreactor’s entire volume can support cell expansion is a great boon when compared to the limited surface area provided by planar platforms, allowing for a much more efficient spatial distribution of cells [67]. The inherent scalability of bioreactors also means that when transitioning to larger production scales a single bioreactor of greater volume can be used instead of increasing the overall number of culture platforms. Because of this, some of the reviewed BEMs were developed so as to validate from a technical and economic perspective the shift of stem cell production methodology from planar platforms to bioreactors [43,46,54].

Karnieli and coworkers described a simple BEM which was used to compare 2D and 3D culture systems in the context of large-scale manufacture of allogeneic MSCs [43]. Specifically, it evaluated the economic benefits of packed-bed bioreactors over a range of production scales. The culture medium was identified as the major cost driver for 2D systems, mostly because of the inclusion of serum in its composition. The use of packed-bed bioreactors led to a 40% reduction of medium consumption for all scales, thus significantly reducing the total bioprocess cost. For larger scales, packed-bed bioreactors also presented considerably smaller requirements in terms of the number of operators and the facility cleanroom footprint, reducing costs even further. The main advantages of packed-bed bioreactors over 2D systems were identified as being the improved surface to volume ratio and the possibility of implementing perfusion feeding. The obtained results demonstrated the need of transitioning MSC culture to bioreactor-based systems during the early stages of development of CTPs to achieve cost efficiency.

Hassan et al. developed a BEM with the purpose of determining the optimal timing of a switch from planar expansion technologies to single-use bioreactors during the development lifecycle of a generic allogeneic MSC therapy product [46]. To achieve this, they combined their previous cost of goods model with a cost of development model and a development lifecycle cash flow model. Commercial scales of 10,000, 500,000 and 100,000 patients were evaluated. For all of the considered scenarios, trading planar technologies for single-use bioreactors was observed to be beneficial, independently of the timepoint when the switch occurred. It was found that switching to single-use bioreactors early during Phase I clinical trials or after the postapproval stage are the most favorable options from an economic perspective, while switching during Phase II and III is relatively inadvisable. However, since making a switch only after the postapproval stage is prone to a higher degree of variability when it comes to out of pocket expenses, switching during Phase I was considered to be the most sensible course of action.

Pinto and coworkers established a BEM in order to evaluate the cost-effectiveness of transitioning from T-175 flasks to vertical-wheel bioreactors (VWBRs) in the context of allogeneic MSC manufacture [54]. Two distinct cell sources were considered, namely MSCs isolated from umbilical cord matrix (UCM) and adipose tissue (AT). When using VWBRs instead of T-175 flasks it was found that the cost of each MSC dose decreased from $17,000 to $11,100 for UCM MSC expansion and from $21,500 to $11,100 for AT MSC expansion. This decrease was made possible mostly because of the significantly reduced labor and quality control costs associated with VWBRs. A further reason for this was that, although the number of batches produced per donor was reduced, due to the high seeding density required to inoculate VWBRs, the number of cells obtained per batch was increased. This resulted in a net increase of the number of doses produced per donor. The greater number of doses meant that the production costs were more spread out and thus the cost per dose was reduced.

### 5.4. Review of Models Used for the Comparison of Manual and Automated Manufacturing

One of the several considerations that should be taken into account when designing stem cell-related bioprocesses is the implementation of automation or semiautomation, which has the advantage of reducing labor, avoiding microbial exposures and increasing product quality consistency due to the reduction of human error [43]. However, automated bioprocessing also typically results in increased FCI costs comparatively to manual bioprocessing, only being more cost-effective than the latter starting from a certain production scale. The BEMs applied to the case studies mentioned below proved to be invaluable tools for assessing the impact of switching from manual to automated bioprocessing in several scenarios, helping to pinpoint the production scale at which an automated process becomes more cost-effective than a manual process [35,45,57].

Jenkins et al. applied a BEM to the design of a bioprocess with the purpose of producing patient-specific hiPSC-derived neurons for drug screening [45]. The model sought to evaluate the economic competitiveness of several planar expansion technologies over a range of production scales, either with manual or automated bioprocessing. A brute-force search algorithm that computed the cost of all possible technology configurations was used to identify the process design which minimized the CoG for each scenario. The main process cost driver was found to be the number of iPSC expansion stages required. Additionally, Monte Carlo simulation was employed to introduce stochasticity into the model and assess the robustness and reproducibility of manual versus automated bioprocessing. It was concluded that process automation reduces the CoG comparatively to manual processes at larger scales, given that the additional indirect costs associated with automation are spread across several cell lines and labor costs are significantly reduced in relation to manual processes. Moreover, automated bioprocesses were found to present less variation in terms of the CoG.

Harrison and coworkers presented a BEM that they used to compare the CoG of large-scale allogeneic MSC production through manual manufacturing in monolayer culture with the CoG of doing the same through SelectT (Sartorius Stedim) automated platforms [35]. Operator labor was found to be a major cost driver due to training expenditures. The implementation of automation significantly reduced the costs associated with operator training even when maintaining almost the same number of operators. In addition to reduced CoG, automation led to an enhanced reproducibility of the final product.

Nießing et al. developed a BEM for comparing the profitability of producing iPSCs in a StemCellFactory (SCF) unit, a fully automated platform developed by the Fraunhofer Institute for Production Technology, with the profitability of manual iPSC production [57]. The SCF platform is claimed to enable improved reproducibility and to increase cell quality by applying standardized procedures that reduce the problems associated with human error, operator-to-operator variability and culture contamination. The comparison of manual and automated iPSC production was performed using three economic analysis metrics, namely return on investment (RoI), payback period and net present value (NPV), measured within a period of 8 years. The metrics showed that automation has better economic potential than manual production, in addition to leading to a higher reproducibility of cell products and an increased throughput. For example, the NPV, which quantifies how much value is added by a project, is far superior for SCF production, nearly $2 million higher, than for the manual production scenario.

### 5.5. Review of Models Used for the Comparison of Culture Media Formulations

Culture medium frequently has a large financial impact on the CoG of stem cell-related bioprocesses and should thus be carefully selected. Regulatory agencies recommend the use of chemically defined and xenofree formulations for prospective clinical applications, so as to ensure product consistency and quality by avoiding the transfer of xenogeneic contaminants to cells and reducing biological variability [15]. Moreover, the expansion rate of cells can differ significantly depending on the chosen formulation, with media that present better performances substantially reducing the resources allocated to produce a certain amount of cells [35]. Therefore, a couple studies have focused on the economic impact of different media formulations for stem cell culture [35,53]. These studies are further described in this subsection.

Harrison and coworkers, whose BEM was already mentioned in the previous subsection, also sought to evaluate the technical and economic impact of using serum-containing medium (SCM) versus serum-free medium (SFM) [35]. In order to have access to realistic input parameters for their modeling, they analyzed the growth kinetics of MSCs from three different donors in both media formulations and found that batch variability was reduced in cells cultured under SFM. From these experimental results, their BEM estimated that the maximum production for a facility with a given capacity was two times higher when SFM was employed. As a consequence, not only are the CoG/dose of MSCs produced with SFM much lower than the CoG/dose of MSCs produced with SCM, but the CoG range is also narrower due to better reproducibility between differing donor cells.

Bandeiras et al. studied the impact of using fetal bovine serum (FBS), an animal-based culture medium, versus human platelet lysate (hPL), a xenofree formulation, on the isolation and expansion of autologous MSCs [53]. A dynamic model using discrete-event simulation was applied to account for the impact of time-dependent operations on the CoG. Additionally, Monte Carlo simulation was carried out in an attempt to represent the biological variability caused by differing donor cells and divergent growth rates for the two culture media. For the analyzed case study, it was found that the harvesting yield (i.e., the number of cells per unit area at confluence) was the most critical parameter when it came to maintaining the cost-effectiveness of the process. For the majority of simulated donors, a lower number of passages was required to produce the target number of cells when hPL was used, which also reduced overall culture time. The lower number of passages required by hPL was due to the smaller size of MSCs grown in this medium, which in turn led to a higher number of cells after MSC isolation from donors. Consequently, although hPL medium costs twice as much as FBS, it provided a more cost-effective alternative in the considered scenario because of reductions in infrastructure and labor costs brought about by the shortened culture time.

### 5.6. Review of Models Used for Health Technology Assessment

Some of the stem cell-related bioprocess models found in the literature have combined bioprocess economic modeling with early health technology assessment, taking into account both clinical trial and manufacturing process data with the intent of determining the market viability of a novel therapy to treat a certain condition [32,51,55]. These studies, analyzed further in this section, resort to disease state transition models to simulate the health progression of patients after being treated. Disease state transition models consist in defining the possible disease states a patient can find himself in, along with the probabilities of a patient in a given state transitioning to any of the other states with the passing of time. The cost-effectiveness of the novel therapy in question is usually compared with the cost-effectiveness of a traditional therapy, in order to assess to which extent the health benefits of the novel therapy outweigh its cost. This can be evaluated by computing the incremental cost-effectiveness ratio (ICER) for each simulated patient. The ICER is the ratio between the difference in costs (comprising medical and manufacturing costs) and the difference in total quality-adjusted life-years (QALYs) relative to both therapies. This metric is then compared to a willingness to pay (WTP) threshold, corresponding to the value society is willing to pay to reduce disease severity [55].

Two seperate research groups focused on assessing the cost-effectiveness of employing implantable devices containing either hESC-derived or hiPSC-derived beta cells for treating type 1 diabetes instead of intensive insulin therapy (IIT) [51,55]. There are some differences between the studies, such as the fact that the first study considered filling the devices with pancreatic progenitors [51], whereas the second study considered filling them with terminally differentiated beta cells [55], making the manufacturing process longer and more complex in the latter case. Despite the differences, both studies stressed the importance of large-scale production to drive down costs per dose and increase the affordability of the new therapy. Bandeiras et al. identified downstream processing as the key limiting factor for process scaling-up, due to the low yields that limit batch size. Wallner et al. showed that hESC-derived beta cell therapy could be cost-effective in comparison to IIT at a WTP threshold of $100,000 per QALY, while Bandeiras et al. found that the therapy could become cost-effective for most of the simulated patients at WTP thresholds between $50,000 and $150,000, as long as manufacturing costs are reduced through scaling-up strategies and medical complications are avoided.

Bandeiras and coworkers also explored the cost-effectiveness of using an MSC-based therapy together with modulators to treat cystic fibrosis instead of a standard therapy with just modulators [32]. Different cell dose numbers, administration regimes (inpatient versus outpatient) and potential increased effectiveness over standard therapy were evaluated. The simulated population showed significant variability in terms of incremental QALYs and the average incremental cost of the cell therapy was high (reaching over $2 million), which prevented the widespread effectiveness of this therapy at typical WTP thresholds ($25,000–150,000/QALY). Increasing the effectiveness of MSC-based therapies in treating cystic fibrosis and reducing manufacturing costs would be necessary to ensure their cost-effectiveness for a wide range of individuals at typical WTP thresholds.

### 5.7. Overview of Model Characteristics

As mentioned in Section 3, bioprocess models can be classified according to three main categories as either static or dynamic, deterministic or stochastic and data-driven or mechanistic. When analyzing the model characteristics of the reviewed stem cell-related BEMs (see respective column of Table 2), it can clearly be observed that the majority of these are static, deterministic and data-driven.

Regarding the first category, only 4 of the reviewed models can be classified as dynamic. This is most likely explained by the much greater complexity of dynamic models and the fact that their results can be more difficult to interpret. Moreover, given that most of the simulated bioprocesses are either nonexistent or still in the earliest stages of development, there is limited information available concerning the logistics of their time-dependent operations. For instance, the biological variability of cell donor samples must be well known and characterized in order to optimize equipment occupation and reduce the additional operating and equipment costs that may arise from suboptimal equipment utilization. As a consequence of this lack of process knowledge, most models follow a static approach, which is still useful for providing early cost estimations and identifying the main cost drivers of bioprocesses. Nevertheless, dynamic models should be constructed whenever possible, since they provide more realistic cost estimates by capturing the intricacies of time-dependent operations.

Concerning the second category, 9 out of the 19 reviewed models are stochastic, with Monte Carlo simulation being by far the most commonly applied stochastic approach. The integration of stochasticity into stem cell-related BEMs is particularly important to evaluate the impact of biological variability on process costs and make risk-based decisions. However, in order to do so, a certain degree of process knowledge is required to understand the ranges of the relevant biological parameters. This knowledge can only be acquired after preliminary experimentation is carried out, and ideally, stochastic simulation should be founded on sufficiently comprehensive data sets. Notably, nearly all of the works which incorporate stochastic elements find themselves resorting to somewhat arbitrary and unrealistic probability distributions, with triangular distributions being extensively employed. Sometimes, these distributions are not even based on previous experiments but rather on subjective expert opinion. This is clearly indicative of the absence of the required knowledge. On the other hand, it should be noted that, with the purpose of testing the influence of parameter variability on model outputs and making more robust conclusions during early process development, deterministic models are usually employed for sensitivity analysis by manually varying the model inputs by a certain percentage. This, in conjunction with the attempts so far made to create stochastic BEMs, demonstrates that the stem cell bioprocess modeling community is well aware of the impact that biological variability can have on the manufacture of CTPs.

Finally, with respect to the third category, only 3 of the reviewed models were found to be mechanistic or semimechanistic. Furthermore, it is telling that both of the fully mechanistic models simulate the production of RBCs or their precursors from HSCs and the semimechanistic model simulates the production of EVs. Erythropoiesis, the process by which HSCs mature into RBCs, is relatively well understood and straightforward, while the semimechanistic model limited itself to establishing an experimentally validated equation for relating the number of EVs produced to the number of cells in culture. In contrast, while much has been learned over recent years, the biological mechanisms underlying stem cell expansion and differentiation are still far from being fully grasped. This is hardly surprising since these mechanisms are extremely difficult to describe, being dependent on an intricate network of several interconnected factors, among which are the extracellular environment, soluble factors and cell–cell interactions. Thus, it is natural that most models follow a data-driven approach to translate these poorly understood phenomena.

## 6. Conclusions

As has been shown by the works here reviewed, BEMs present enormous potential as decision support tools for CTP bioprocess development. They can be used to readily evaluate the impact of process modifications on the CoG, identify limiting technologies and process steps on which further improvement should be focused and determine optimal bioprocess configurations for a multitude of scenarios. The in silico estimations that BEMs provide, which are obtained in a swift and inexpensive manner, are thus capable of guiding a manufacturer’s decision-making and significantly reducing the costs of process discovery and development. Because of this, BEMs can play a pivotal role in the QbD framework. They are ideal platforms for defining and exploring the design space of CTPs throughout the various stages of iterative refinement, making them especially relevant for developing bioprocesses that must adhere to rigorous CQAs in order to obtain regulatory approval.

At the same time, the reviewed studies also demonstrate that it is necessary to obtain a deeper understanding about the influence of critical biological parameters on product CQAs. This knowledge, which should come from meticulously planned and conducted experimentation, is of paramount importance to further refine BEMs, improving them from more simplistic models, i.e., static and deterministic models, to more complex and reliable models, i.e., dynamic and stochastic models. Only in this way will it be possible to construct high-fidelity models that faithfully represent the dynamic nature and inherent variability of stem cell-based manufacturing. These robust BEMs may then be employed for the careful examination of diverse manufacturing scenarios, facilitating the incorporation of innovative strategies and technologies. In turn, this would increase and accelerate the market acceptance of the CTPs associated with their simulated bioprocesses, which at present are often held back from regulatory approval or give rise to unforeseen prohibitive costs because of failed batches caused by a lack of process knowledge and characterization.

## Figures and Tables

**Figure 1 bioengineering-09-00791-f001:**
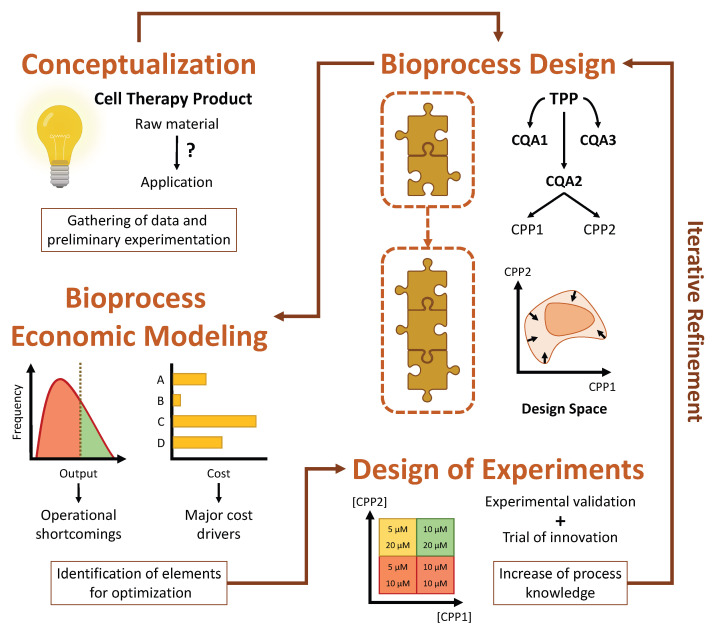
In the context of the quality by design framework, the development of a cell therapy product (CTP) begins with the conceptualization of a bioprocess that leads from cell-based raw materials to a product capable of fulfilling its envisaged application. It then moves on to the creation of a rough draft of the bioprocess, defining the CTP in terms of its target product profile (TPP), critical quality attributes (CQAs) and critical process parameters (CPPs). Based on the CPPs, a process design space is also delineated. Naturally, the initial rough draft is limited by a lack of process knowledge, but it provides a blueprint for future development. So as to guide the accrual of process knowledge, a bioprocess economic model (BEM) can be established to simulate the outlined bioprocess, serving as a useful tool for swiftly and inexpensively identifying those elements whose optimization would bring the most benefit to the bioprocess in terms of quality or cost. Once these have been identified, directed experimentation should be carried out following a methodical design of experiments approach. The results of this experimentation effectively increase process knowledge, which can then be used to improve the initial bioprocess design, clarifying its TPP and narrowing its design space. The established BEM should then also be improved, increasing its fidelity, and used to identify further elements for optimization. This process, termed iterative refinement, is cyclically repeated, until a robust and economically viable bioprocess is achieved.

**Table 1 bioengineering-09-00791-t001:** Summary of the advantages and disadvantages of bioprocess model characteristics.

Model Characteristic	Advantages	Disadvantages
Static	Simple and easy to build Apt for early process development	Disregard resource constraints Ignore time dependence of operations
Dynamic	Can describe realistic workflows Capture intricate logistical operations	Complex and arduous to build Sometimes are difficult to interpret
Deterministic	Easy to interpret Allow for sensitivity analysis	Do not take into account process variability
Stochastic	Incorporate uncertainty Can be used for risk-based analysis Convey the statistical robustness of outputs	Require information regarding the variability of inputs Their interpretation requires intricate statistical analysis
Mechanistic	Can describe complex phenomena	Require a large amount of knowledge
Data-driven	Transmit relevant information in a simple manner	Can predict phenomena but never explain them

**Table 2 bioengineering-09-00791-t002:** Summary of studies which report stem cell-related bioprocess economic models. This list attempts to be comprehensive, covering multiple stem cell types, case studies with a diverse set of objectives and models with various combinations of the core characteristics previously described, but is not guaranteed to be entirely exhaustive. The works are listed in chronological order.

Cell Type	Case Study	Model Characteristics	Study
Allogeneic MSCs	Optimization of cell expansion technologies for the large-scale expansion of MSCs in various scenarios.	Static, Deterministic, Data-driven	[41]
Allogeneic hiPSCs	Optimization of bioreactor type and purification technologies for the large-scale generation of hiPSC-derived cardiomyocytes.	Static, Deterministic, Data-driven	[42]
Allogeneic MSCs	Comparative evaluation of 2D versus 3D cell expansion technologies for the large-scale expansion of MSCs in various scenarios.	Static, Deterministic, Data-driven	[43]
Allogeneic MSCs	Optimization of downstream processing and fill finish technologies for the large-scale expansion of MSCs in various scenarios.	Static, Deterministic, Data-driven	[44]
Autologous hiPSCs	Optimization of 2D cell expansion technologies for the generation of hiPSC-derived neurons for drug screening in various scenarios. Comparative evaluation of manual versus automated manufacturing for this same bioprocess.	Static, Deterministic/Stochastic ^1^, Data-driven	[45]
Allogeneic MSCs	Impact of switching from 2D cell expansion technologies to microcarriers in SUBs at different stages of development of MSC-based cell therapy products in various scenarios.	Static, Stochastic, Data-driven	[46]
Autologous hiPSCs	Comparative evaluation of affinity purification technologies for the generation of hiPSC-derived progenitor photoreceptors.	Static, Deterministic, Data-driven	[47]
HSCs	Optimization of the production of HSC-derived red blood cells in a dual hollow-fiber bioreactor.	Static, Stochastic, Mechanistic	[48]
Allogeneic MSCs	Comparative evaluation of cell expansion technologies for the production of MSC-based cell therapy products in various scenarios.	Dynamic, Stochastic, Data-driven	[49]
Allogeneic MSCs	Comparative evaluation of cell expansion technologies for the large-scale expansion of MSCs derived from umbilical cord matrix for the treatment of acute graft-versus-host disease.	Static, Stochastic, Data-driven	[50]
hESCs	Health technology assessment of implantable devices containing hESC-derived beta cells for treating type 1 diabetes.	Static, Stochastic, Data-driven	[51]
HSCs	Optimization of erythroblast production in STBRs through optimum media exchange time, cell density and cell productivity.	Static, Deterministic, Mechanistic	[52]
Allogeneic MSCs	Comparative evaluation of SCM versus SFM for the expansion of MSCs in 2D cell expansion technologies. Comparative evaluation of manual versus automated manufacturing for this same bioprocess. Impact of donor variability, quality control and product transport on this same bioprocess.	Dynamic, Stochastic, Data-driven	[35]
Autologous MSCs	Impact of using hPL (xeno-free) versus FBS (animal-based) culture media for the isolation and expansion of MSCs.	Dynamic, Stochastic, Data-driven	[53]
Allogeneic MSCs	Impact of using microcarriers in VWBRs versus planar culture platforms for the expansion of MSCs.	Static, Deterministic, Data-driven	[54]
Allogeneic hiPSCs	Health technology assessment of implantable devices containing hiPSC-derived beta cells for treating type 1 diabetes.	Static, Deterministic ^2^, Data-driven	[55]
Allogeneic MSCs	Health technology assessment of an MSC-based therapy for reducing inflammation in cystic fibrosis patients.	Dynamic, Stochastic, Data-driven	[32]
Allogeneic MSCs	Optimization of cell expansion and EV harvest technologies for the large-scale production of EVs by MSCs in various scenarios.	Static, Deterministic, Semi-mechanistic	[56]
Autologous hiPSCs	Comparative evaluation of manual versus automated manufacturing for the expansion of hiPSCs in 2D cell expansion technologies.	Static, Deterministic, Data-driven	[57]

Abbreviations. DS, dextran sulfate; EVs, extracellular vesicles; FBS, fetal bovine serum; hESCs, human embryonic stem cells; hiPSCs, human-induced pluripotent stem cells; hPL, human platelet lysate; HSCs, hematopoietic stem cells; MSCs, mesenchymal stromal cells; SCM, serum-containing media; SFM, serum-free media; STBRs, stirred-tank bioreactors, SUBs, single-use bioreactors; VWBRs, vertical-wheel bioreactors. ^1^ This study reports a deterministic bioprocess model for the optimization of 2D cell expansion technologies and a stochastic bioprocess model for the comparative evaluation of manual versus automated bioprocessing strategies. ^2^ While the bioprocess model reported in this study is deterministic, it is integrated with a stochastic disease state transition model.

## Data Availability

Not applicable.

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
