# Peer review of "Bioprocess Economic Modeling: Decision Support Tools for the Development of Stem Cell Therapy Products"

_bioengineering, 2022, doi:10.3390/bioengineering9120791_

Round 1

Reviewer 1 Report

After reading such an attractive title, the text has not generated answers and unfortunately has generated doubts about the different aspects involved in the economic aspects of the production of drug therapy drugs.

One aspect to consider is that different types of cells need different types of culture and in each case there are numerous specific economic aspects to evaluate. For example, MSCs are easy to scale up their culture process and the qualification of personnel as well as the time and complexity of the procedure is less than for the isolation and culture of iPSCs and subsequent pre-differentiation.

Among the aspects included that influence the cost/quality of the product, there are aspects that need to be evaluated and are critical at an economic level: Donor/initial yield, the difference between donors when scaling up a cell culture influences the rest of the parameters; Product shelf life, the most cost-effective, large-scale production is difficult in live drugs due to their rapid shelf life, you could evaluate this aspect and even propose, if you consider it appropriate a large-scale production up to an intermediate cell product that can be cryopreserved; And finally the training aspects of the personnel involved as well as the costs of product release (quality and safety testing, pharmacovigilance) and the costs of transporting the final product. Please analyse these aspects. 

Reviewer 2 Report

This is a review that aimed to overview recently published studies about the enormous potential of bioprocess economic models (BEMs) which were used as tools to guiding decision-making in bioprocess design, supporting process optimization while also helping to reduce the overall production costs in cell and gene therapy related bioprocess. In addition, some theoretical concepts relevant to the building and structuring of BEMs are explored, including different bioprocess model, capital investment (FCI), cost of goods (CoG), cost of quality (CoQ), and so on. In general, the topic of this manuscript is important which may have certain guiding significance for the future study of stem cell therapy for clinical translation. Some suggestions are list as follows.  

1. The author should give a detailed explanation of the BEMs. By using such tool, how to make sure that the steps involved in acquisition of generation of stem cell product are economically viable.  

2. A frame diagram for the structure of the whole article is needed to help the readers better understand the content of the review.  

3. In the sub-section of Part 5, most of the content are not consistent with the title. For example, in sub-section 5.2, the tile is Optimization of Bioprocess Design. However, there is no description about how to make sure the optimization.

Reviewer 3 Report

The manuscript presents a very interesting overview over different economic models of cell therapy processes. The paper fits into the scope of the Journal and is really interesting for everyone who wants to implement models to evaluate biotechnological processes economically. The text is written well except some small lapses but can be structured more purposefully.

The publication should be accepted after minor revision. The following comments might help to revise.

Page 2 line 52: “stark” possibly too colloquial

Page 2 line 68-71: This description for bioprocess models is not very well done.

Page 3 line 107: The comma at the end of the line is to much.

Page 3 line 109-114: This sentence is too long and therefore confusing

Page 7 table 1: This table is very confusing, please use keywords rather than sentences

Page 9 table 2: Please structure this table more clearly. Is it really relevant for this work which focus the different studies had? Wouldn't it be more helpful to show the models used or sorted by cell type?

Page 9 table 2 following: Regarding the title of the paper "Bioprocess economic modeling" and "decision support tools", a different structuring would be helpful. I think that the papers should be discussed in a differentiated way not on the basis of the biological results but on the basis of the models used. What distinguishes the models from each other, what are strengths and weaknesses, what are the input parameters and what is the respective model output.

Round 2

Reviewer 1 Report

Thanks for your answer. According to his comments, the paper subtly deals with different aspects in order to give the readers a first idea of the production systems and associated costs in cell therapy, obviously without going in depth into the different systems. With this in mind, the expanded aspects have improved understanding.